# An Ontology-Based Recommender System with an Application to the Star Trek Television Franchise

**Paul Sheridan [1,*]** , **Mikael Onsjö [2]** , **Claudia Becerra [3]** , **Sergio Jimenez [4]** **and George Dueñas [4]**

[1] Tupac Bio, Inc., San Francisco, CA 94103, USA
[2] Independent Researcher, London, SE13 7NZ, UK
[3] Systems and Computer Engineering Department, Universidad Nacional de Colombia, Ciudad Universitaria bldg. 453, Bogotá, D.C. 11001, Colombia
[4] Insitituto Caro y Cuervo, Calle 10 # 4-69, Bogotá, D.C. 111711, Colombia
[*] Correspondence: paul.sheridan.stats@gmail.com

**Abstract:** Collaborative filtering based recommender systems have proven to be extremely successful in settings where user preference data on items is abundant. However, collaborative filtering algorithms are hindered by their weakness against the item cold-start problem and general lack of interpretability. Ontology-based recommender systems exploit hierarchical organizations of users and items to enhance browsing, recommendation, and profile construction. While ontology-based approaches address the shortcomings of their collaborative filtering counterparts, ontological organizations of items can be difficult to obtain for items that mostly belong to the same category (e.g., television series episodes). In this paper, we present an ontology-based recommender system that integrates the knowledge represented in a large ontology of literary themes to produce fiction content recommendations. The main novelty of this work is an ontology-based method for computing similarities between items and its integration with the classical Item-KNN (K-nearest neighbors) algorithm. As a study case, we evaluated the proposed method against other approaches by performing the classical rating prediction task on a collection of Star Trek television series episodes in an item cold-start scenario. This transverse evaluation provides insights into the utility of different information resources and methods for the initial stages of recommender system development. We found our proposed method to be a convenient alternative to collaborative filtering approaches for collections of mostly similar items, particularly when other content-based approaches are not applicable or otherwise unavailable. Aside from the new methods, this paper contributes a testbed for future research and an online framework to collaboratively extend the ontology of literary themes to cover other narrative content.

**Keywords:** knowledge-based recommender systems; knowledge representation; literary theme; ontological engineering; ontology population; ontology-based recommender systems; Star Trek

## 1. Introduction

Recommender systems (RSs), or recommenders for short, help users to navigate large collections of items in a personalized way [1]. Broadly speaking, RSs function by correlating user preference data with item attribute data to generate a ranked list of recommended items for each user. Systems of this kind play a crucial role in our modern information overloaded society. RSs are linked to the area of affective computing and sentiment analysis [2], as they combine user opinions and feelings to produce predictions in affective dimensions, including (but not limited to) such dimensions as 'like' and 'dislike'.

RSs have received special attention in the television domain over the last two decades (see [3] for an extensive survey), due in part to a rapid increase in the production of scripted TV series in conjunction with an online streaming service boom [4]. One issue concerning scripted TV series is that, in most RSs, items are considered at "channel" or "program" level [3]. As a consequence, scripted TV series consisting of dozens or even hundreds of episodes get treated as single items [5]. In addition, RSs applied to the television domain face the general item cold-start problem [6], as well as the particular issue of scripted TV series consisting of sets of mostly similar items (i.e., episodes) that are difficult to differentiate among for the purpose of recommendation. We address these issues in the present work by enriching the representation of scripted TV episodes through the use of an ontology of literary themes.

The cold-start problem refers to the temporal situation where there is not enough information for an RS to produce new item or user recommendations. The user-cold-start scenario is commonly addressed by constructing a user profile by appealing to explicitly or implicitly provided user information (e.g., user preferences, demographic information, browsing history) [7–9]. In this paper, we address the item-cold-start scenario, where it is necessary to item metadata to associate it with other items that have already been seen by users. In contrast, the warm-system scenario is the ideal situation when both items and users are already known by the system, providing the conditions to perform a static evaluation. In our evaluations, we use item-cold-start (hereinafter cold-start) and warm-system settings to compare the performance of different recommendation algorithms.

Star Trek stands out among television series for its cultural significance, number of episodes, and longevity [10] (STARFLEET, The International Star Trek Fan Association, Inc. (2019) [11]). For these reasons, we elected to use Star Trek as a testbed to develop and evaluate the RSs presented in this paper. Another reason motivating our choice of Star Trek is the availability of multiple sources of information, including transcripts, user ratings, sets of tags associated with episodes, and an ontology hierarchy of tags. In particular, we used a detailed ontology of literary themes [12,13] that was initially developed for Star Trek but has since evolved to be used for general works of fiction. This particular situation allowed us to carry out a thorough evaluation that produced insights about the usefulness of different information resources for the construction of RSs, as well as a means to evaluate a new approach for RSs based on ontology.

Most state-of-the-art RSs are built using collaborative filtering (CF) [1,3,6], which is based solely on the analysis of user assigned item preferences. Common CF methods include the classical $k$-nearest neighbors algorithm [14], non-negative matrix factorization [15], and recent approaches based on efficient representations obtained using deep learning [16]. The CF approach is popular because of its high performance and independence from the nature of the items. However, it is particularly weak against the issues mentioned for the TV series domain. As an alternative, content-based filtering (CBF), knowledge-based filtering (KBF), and ontology-based filtering (OBF) approaches aim to leverage available domain knowledge in an effort to speed up the discovery of relationships between items and users, which CF approaches require large amounts of data to infer [17].

The most common role of ontology in RS design consists of providing a taxonomic classification of items [18–20]. According to this approach, user profiles are indexed by the entities from the ontology and each dimension is weighted according to the preferences of the explicit (e.g., ratings, likes) [18,19] and/or implicit users (e.g., reads, views) [20]. Non-domain ontologies have also been used in combination with domain ontologies to aid the recommendation process, but their integration is achieved by means of rules or inference engines [21–23]. Other approaches aim to model both users and items with ontologies looking for relationships in a single semantic space [22,24,25]. Most of these approaches either use lightweight ontologies or use ontologies as a source for a controlled vocabulary, resulting in a shallow or null use of the ontological hierarchical structure. The exceptions are the approaches that use inference engines [18,23]. Recent approaches [26,27] manage to handle large ontologies while exploiting relationships along the entire hierarchy.

When practitioners plan to develop a new RS, they face the problem of selecting those information resources needed to bootstrap the new system. During these initial stages, CBF/KBF/OBF approaches are to be preferred over CF alternatives, given the obvious lack of user feedback. Although there is an extensive body of literature on each of these approaches, traversal studies comparing the performances of alternative methods on a single dataset are scarce. In this paper, we aim to fill this gap and to provide practitioners with useful insights for decision-making regarding information resources.

We focus on the OBF approach because this is the one where the results depend mainly on developers and because ontologies are the most informative resource to start a new RS. For instance, CBF approaches depend heavily on the nature of the items (e.g., books, hotels, clothing, movies), whose native data representations are not always RS friendly. Similarly, CF approaches depend on the number of users, which in turn depends on external factors such as popularity, advertising, trends, etc. However, there are many factors in the construction of an ontology for an RS that influences recommendation quality. One important issue pertains to the amount of domain knowledge that can be encoded in the ontology. Many domains provide only shallow ontologies that are unable to leverage an RS. This is the case for the episodes of television series such as Star Trek, which with some effort can be organized into a taxonomy with a few dozen classes. In this paper, we present a method to exploit a subordinate ontology of literary themes that models features of the episodes instead of the episodes themselves. Our hypothesis is that, if a much more detailed and non-domain specific ontology is available, then a recommendation engine can exploit it to produce higher quality recommendations.

Another issue arising when developers engage in the construction of an ontology is that most existing studies do not provide information about the insights on ontology development and the item annotation process. In what follows, we provide non-technical descriptions of the motivations behind the presented ontology and a detailed example of the process of annotating a Star Trek episode with literary themes drawn from the ontology (see Appendix A). In addition, aside from the mandatory quantitative evaluation of the presented methods, we provide a proof of concept by means of a qualitative assessment of the neighbor episodes of a particular Star Trek episode in a web application implemented with the resources proposed in this paper (see Appendix B).

The rest of the paper is organized as follows. In Section 2, we describe the materials used and our proposed methodology. In Section 3, we evaluate our method by predicting the preferences (i.e., ratings) given by a set of users to a set of items (i.e., Star Trek episodes) in the classical rating prediction task (Warm System) and in the item cold-start scenario (Cold Start). In Section 4, we discuss the results and share some concluding remarks. Finally, in the appendices, we look at an example episode to illustrate the system of thematic annotation we employed and show how our R Shiny web application can be used to recommend Star Trek television series episodes. Data and computer code availability is described in Supplementary Materials.

## 2. Materials and Methods

### 2.1. Neighborhood-Based Collaborative Filtering

An RS is an algorithm that associates a set of items and users providing a ranked list of items to each user according to their preferences [28,29]. Formally, items and users are represented in a ratings matrix, $\mathbf{R} = \{r_{u,i}\}$, where $r_{u,i}$ is the degree of preference of user $u$ to item $i$. The task of the RS is to assign preference predictions, $\hat{r}_{u,i}$, to all missing values in the (usually sparse) matrix $\mathbf{R}$. CF recommendation is a strategy to obtain those predictions by exploiting item–item and user–user relationships under the hypothesis that similar users ought to prefer similar items.

The most popular methods for addressing CF consist of reducing the dimensionality of $\mathbf{R}$ using matrix factorization (MF) [15], singular value decomposition (SVD) [30], and other related techniques [31–33]. Another approach consists of using the information of the $k$-nearest neighbor items or users to make rating predictions. The most representative method of this approach is the Item-KNN algorithm as proposed by Koren [14]. This method makes use of a similarity function, $S$, that provides

a similarity score, $s_{i,j}$, for any pair of items $i$ and $j$. This function, denoted by $S^k_{i,u}$, is used to identify the set of $k$ items rated by a user $u$ being most similar to an item $i$. Formally,

$$\hat{r}_{u,i} = b_{u,i} + \frac{\sum_{j \in S^k_{i,u}} s_{i,j}(r_{u,j} - b_{u,j})}{\sum_{j \in S^k_{i,u}} s_{i,j}}. \tag{1}$$

This model is adjusted by baseline estimates $b_{u,i}$ representing the rating bias of the user $b_u$ that of the item $b_i$, and the system overall bias $\mu$ (i.e., the mean of the ratings in **R**). The bias $b_{u,i}$ is simply $\mu + b_u + b_i$; and $b_u$ and $b_i$ are computed as follows:

$$b_i = \frac{\sum_{u \in U_i}(r_{u,i} - \mu)}{\lambda_1 + |U_i|} ; b_u = \frac{\sum_{i \in I_u}(r_{u,i} - \mu - b_i)}{\lambda_2 + |I_u|}. \tag{2}$$

Here, $U_i$ is the set of users who rated $i$, $I_u$ is the set of items rated by $u$, and finally $\lambda_1$ and $\lambda_2$ are regularization parameters determined by cross validation. In summary, $\hat{r}_{u,i}$ is a weighted average of the ratings of the $k$-most similar items to $i$ rated by $u$ considering the item and user bias, which are their respective mean deviations from the average.

The flexibility of this model relies on the item-similarity function $S$, which can be built with any informational resources at hand. The natural choice for $S$ is to obtain it from correlations between the items in **R**. We refer to this CF model as IKNN (item K-nearest neighbors). Note that this choice is particularly weak against the item cold-start problem [9]. When a particular item has not been rated by a considerable number of users, the correlations of its ratings against other items are in most cases non-statistically significant.

*2.2. Item Similarity*

In this paper, we exploit the flexibility of the IKNN model by using different representations of items to build alternatives for $S$. In the following subsections, we present a set of item-similarity functions using three different representations for the items. First, in Section 2.2.1, the items have textual representations, which produce a CBF recommender when used in Equation (1). Second, in Section 2.2.2, the items are represented by sets of tags from a controlled vocabulary (i.e., tags), making the approach a KBF recommender. Finally, in Section 2.2.3, the tags are arranged in a semantic hierarchical structure, converting the previous approach into an OBF recommender. In that subsection, we present a novel approach to compare items based on an ontology which, instead of modeling items, uses the ontology itself to model item features. This strategy allows one to leverage the knowledge of an ancillary ontology when the ontology that models the items is either considerably smaller or non-available.

2.2.1. Textual Representation

In many scenarios, it is possible to obtain a textual representation for the items to be considered by an RS. Such representations can range from short descriptions to complete item representations in the case of textual documents (e.g., books, articles, contracts). The most common approach to build a textual similarity function is to use the vector space model approach [34], which consists of building a word-item matrix from the collection of texts to be compared. The entries of the matrix $\mathbf{M} = m_{w,i}$ are weights associated with the occurrence of the word $w$ in the text associated with the item $i$. The weights $m_{w,i}$ are commonly determined using the well-known term frequency–inverse document frequency *tf-idf* weighting schema

$$m_{w,i} = f_{i,w} \log \frac{N}{n_w}, \tag{3}$$

where the term frequency (TF) $f_{i,w}$ is the number of times $w$ occurs in the textual representation of $i$, $N$ is the total number of items in the collection, and $n_w > 0$ is the number of items in which $w$ occurred

in their textual representations. The inverse document frequency (IDF) $\frac{N}{n_w}$ is the proportion of items in the collection containing the term $w$. Finally, the item similarity score is computed according to the cosine similarity:

$$s_{i,j}^{\text{TFIDF}} = \frac{\sum_{w \in W} m_{w,i} m_{w,j}}{\sqrt{\sum_{w \in W} m_{w,i}^2} \sqrt{\sum_{w \in W} m_{w,j}^2}}. \tag{4}$$

Here, $W$ is the vocabulary of words used in the textual collection. A CBF recommender, which we refer to in our experiments as TFIDF, is produced when the similarity function $s_{i,j}$ in Equation (4) is used in Equation (1).

Another common approach to compare texts is latent semantic indexing (LSI) [35], which addresses the sparsity of **M** due to the usual large size of the vocabulary $W$. This is achieved by factorizing **M** using SVD: $\mathbf{M} = \mathbf{U\Sigma V}^T$. The resulting matrices **U** and **V** are orthogonal and **Σ**, which is diagonal, contains the singular values of the decomposition. By selecting the $p$-largest singular values and replacing the remaining ones by zeros in **Σ**, it is possible to obtain a representation of dimension $p$ for each text that can be used to build a similarity function by using again Equation (4). These $p$-dimensions are known as latent factors. Usually, $p$ is a free parameter to be determined for each data collection and application. In our experiments, we refer to the CBF recommender obtained using this method as LSI-$m$.

### 2.2.2. Content Representation Based on Controlled Vocabularies

A controlled vocabulary is a set of standardized terms used to annotate and navigate content. Content representation based on controlled vocabularies allow for items to be mapped to a semantic set of features. This, in principle, makes for more compact and informative representations as compared with textual representations. Controlled vocabularies come in different flavors. There are controlled vocabularies developed by domain experts [36–38], those extracted automatically from textual representations [8], and those constructed in a collaborative way by groups of users [8,39–41].

Once some items are represented as sets of binary features, an item similarity function can be built using any resemblance coefficients based on cardinality, such as the Jaccard [42] and Dice [43] coefficients. In this context, let $i$ and $j$ be sets of features representing items from a collection of items in an RS. The corresponding item similarity functions are given by

$$s_{i,j}^{\text{JACCARD}} = \frac{|i \cap j|}{|i \cup j|}, \quad \text{and} \quad s_{i,j}^{\text{DICE}} = \frac{|i \cap j|}{2(|i| + |j|)}, \tag{5}$$

respectively.

As before, it is possible to exploit the fact that features occurring many times in an item collection are relatively less informative than less frequent features. It is possible to reuse the IDF factor to take advantage of this fact. The TF factor is not relevant in this scenario because the features are binary, that is, they can occur at most once for each item. The resulting similarity between items $i$ and $j$ can be expressed by the formula

$$s_{i,j}^{\text{COSINE\_IDF}} = \frac{\sum_{w \in i \cap j} \left(\frac{N}{n_w}\right)^2}{\sqrt{\sum_{w \in i} \frac{N}{n_w}} \sqrt{\sum_{w \in j} \frac{N}{n_w}}}, \tag{6}$$

where, again, $N$ is the number of items to be recommended, and $n_w$ the number of items having the feature $w$.

### 2.2.3. Ontology-Based Similarity Functions

In computer science, an ontology is a semantic organization of entities describing a particular domain [44]. The simplest kind of ontology is a tree with root node $\phi$, which represents the most general entity. The structure extends down from the root node to other more specific entities related by the *is-a* relation. Formally, an ontology, $O$, is a tuple, $(E, \phi, p)$, where $E$ is a set of entities, $\phi$ is the

root entity having $\phi \in E$, and $p$ is the parent entity function $p : E \to E$, defined in such a way that each entity has an unique an non-cyclical path to $\phi$. A semantic similarity function on $O$ is a function that computes pairwise similarities between entities [45]. Let $t$ and $u$ be entities in $O$. Most of these functions are built from the following primitives:

$d$: The maximum number of steps from any entity to $\phi$.

$m$: The maximum number of steps between any pair of entities.

$depth_t$: The number of steps from entity $t$ to $\phi$.

$path_{t,u}$: The number of steps from entity $t$ to entity $u$.

$LCS_{t,u}$: The least common subsumer of $t$ and $u$ (i.e., their deepest common ancestor).

$IC_t$: The *information content* (IC) of entity $t$ as proposed by Resnik et al. [46]. *IC* is a measure of the informativeness of an entity in a hierarchy obtained from statistics gathered from a corpus. In our scenario, the corpus is a collection of items, each of which is represented by a set of features associated with entities in an ontology. Thus, the *IC* of an entity $t$ is defined as $IC_t = -\log P(t)$, where $P(t)$ is the probability of $t$ in the corpus. For instance, assume the corpus of items to be a set of TV episodes whose features are themes from a theme ontology. In addition, assume the following path of theme entities in an is-a chain: "wedding ceremony" $\to$ "ceremony" $\to$ "event" $\to \phi$. The probability $P(t)$ of an entity $t$ is the ratio between its number of occurrences $f_t$ and the total number of entity occurrences in the corpus $M$. Each occurrence of the theme "wedding ceremony" increases the counting up the hierarchy until $\phi$ is reached. Therefore, $P(\phi) = 1$ and $IC_\phi = 0$. The IC scores agree with the information theoretical principle that events having low probability are highly informative and vice versa.

Figure 1 illustrates the primitives in the computation of ontology-based similarity functions. There are several mathematical expressions that combine these primitives to produce similarity functions. We consider the following representative similarity functions:

$$
\begin{aligned}
S_{t,s}^{\text{PATH}} &= \frac{1}{path_{t,s}+1} & S_{t,s}^{\text{WUP}} &= \frac{2 \cdot depth_{LCS_{t,s}}}{depth_t + depth_s} \\
S_{t,s}^{\text{RES}} &= \frac{IC_{LCS_{t,s}}}{c_2} & S_{t,s}^{\text{LCH}} &= -\log\left(\frac{path_{t,s}}{2d}\right) \cdot \frac{1}{c_1} \\
S_{t,s}^{\text{LIN}} &= \frac{2 \cdot IC_{LCS_{t,s}}}{IC_t + IC_s} & S_{t,s}^{\text{JCN}} &= \frac{1}{c_3 \cdot (IC_t + IC_s - 2 \cdot IC_{LCS_{t,s}})}.
\end{aligned}
\tag{7}
$$

The function $S_{t,s}^{\text{PATH}}$ is a commonly used conversion of $path_{t,s}$ to a similarity score in the unit interval. Leacock and Chodorow's $S_{t,s}^{\text{LCH}}$ function adjusts the inverse of $path_{t,s}$ to account for the total depth of the ontology [47]. Wu and Palmer's $S_{t,s}^{\text{WUP}}$ function relates the depths in the ontology representing the commonalities between $t$ and $s$ with the Dice coefficient by their least common subsumer [48]. Analogously, Lin's measure uses ICs instead of depths with the same coefficient [49]. Resnik's function $S_{t,s}^{\text{RES}}$ simply uses the IC of $LCS_{t,s}$ [46]. Finally, Jiang and Conrath's function $S_{t,s}^{\text{JCN}}$ is a different formulation of Lin's using the same arguments. The functions $S_{t,s}^{\text{LCH}}$, $S_{t,s}^{\text{RES}}$ and $S_{t,s}^{\text{JCN}}$ produce similarity scores that can be larger than unity. To keep these measures close to the unit interval, the $c_i$'s are used to scale the scores with the values $c_1 = 3$, $c_2 = 10$, and $c_3 = 2$. These functions fulfill to some extent the principle of identity by returning scores close to 1 when the entities are very similar, and scores close to 0 when they have few commonalities.

As in Sections 2.2.1 and 2.2.2, our goal is to build an item similarity function from the similarity functions in Equation (7). As before, the resulting functions for the items to be recommended can be substituted in Equation (1) to produce an OBF recommender method based on ontology $O$. In so doing, we follow the exposition of Jimenez et al. [50], who proposed soft cardinality as a mechanism to compare pairs of sets of items represented as sets of entities making use of a similarity function for

entities. Let the *i* be represented by a set of entities in $O$, $i = \{t_1, t_2, \cdots, t_{|i|}\}$. The soft cardinality of *i* is defined by the formula

$$\langle i \rangle = \sum_{t \in i} \left( \frac{1}{\sum_{s \in i} (S^*_{t,s})^p} \right),$$ (8)

where $S^*_{t,s}$ is any function in Equation (7), and $p > 0$ a softness-control parameter. The intuition underlying soft cardinality is that when the entities representing *i* are very similar among them, then $\langle i \rangle \to 1$; and when the entities are pretty much differentiated, then $\langle i \rangle \to |i|$ (i.e., the classical set cardinality). To compare a pair of items *i* and *j*, it is necessary to obtain $\langle i \cup j \rangle$, which can be computed using Equation (8). For the intersection, it is necessary to make use of the "soft cardinality trick", which consists of inferring the intersection form the soft cardinalities of *i*, *j* and $i \cup j$ by the following expression:

$$\langle i \cap j \rangle = \langle i \rangle + \langle j \rangle - \langle i \cup j \rangle.$$ (9)

The "trick" allows soft cardinality to measure non-empty intersections when $i \cap j$ is the empty set. For instance, using again our running example of TV episode items and thematic entities, if $i = \{\text{"journey", "ceremony"}\}$ and $j = \{\text{"time travel", "wedding ceremony"}\}$, then $|i \cap j| = 0$ while $\langle i \cap j \rangle > 0$ because of the non-zero similarities that can be obtained from the ontology between the elements of *i* and *j*.

The softness control parameter *p* is discussed at length in Jimenez et al. [50]. Suffice it to say here that maximal "softness" is obtained in the limit as *p* approaches $0^+$ (making $S^p_{t,s} \to 1$ for any *t* and *s*), maximal "crispness" is obtained in the limit as *p* approaches $\infty$ (making $S^p_{t,s} \to 1$, if $t = s$, and 0 otherwise), while setting $p = 1$ leaves the values of $S(x, y)$ unmolested. Note that soft cardinality, $\langle \cdot \rangle$, generalizes the set theoretic notion of cardinality defined above to non-whole number values by exploiting pairwise similarities between entities in the calculation of item cardinality. Classical cardinality, $| \cdot |$, by contrast, is confined to the whole numbers.

The cardinality-based coefficient that we employ is the cosine index integrated with soft cardinality:

$$s^p_{i,j} = \frac{\langle i \cap j \rangle}{\sqrt{\langle i \rangle \cdot \langle j \rangle}}.$$ (10)

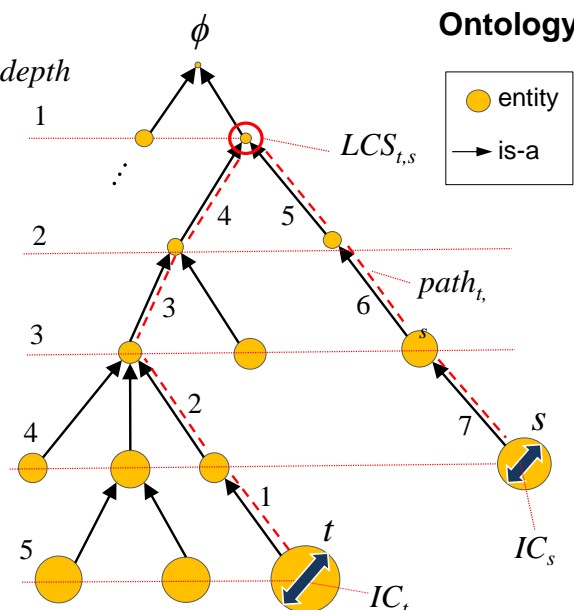

**Figure 1.** Primitives of the entity-similarity functions in an ontology.

The function in Equation (10) is an item similarity function that can be used in Equation (1) to produce an OBF recommender. The argument $p$ represents the entity similarity function used to build the soft cardinality operator $\{\cdot\}$ that is one of the functions in Equation (7). Therefore, this method produces six OBF recommenders that we identify by their inner entity similarity function, namely: PATH, WUP, RES, LCH, LIN, and JCN. Each one of these six recommenders is controlled by the soft cardinality softness controller parameter $p$. Thus, $p$ controls the degree to which knowledge from the ontology hierarchy is used in the RS. When the value of $p$ becomes a relatively large number (e.g., $p = 20$), the effect of the ontology hierarchy is null. This happens because $p$ is used in Equation (8) as the exponent of the similarity scores between the ontology entities. Since these scores are mostly in the unit interval, they become close to 0 when raised to the $p$-power. The other extreme of the values of $p$ is when they approach 0, making the same exponentiations yield values close to 1. This can be interpreted as an overuse of the ontology hierarchy because even a small similarity score between a pair of entities yields to the conclusion that they are practically identical. Therefore, appropriate values for $p$ fall between these extremes. In the experiments presented in Section 3, for each one of the six OBF recommenders, we adjust and report the values of $p$ that obtained the best performance.

### 2.3. A Literary Theme Ontology with Application to Star Trek Television Series Episodes

In practice, the choice of RS depends on the resources available for generating recommendations. The above proposed CBF, KBF, and OBF hybrid RSs cover a wide range of real-world use cases [1,3,51]. To compare the proposed RSs across these three alternatives, it is necessary to have a dataset that all relevant representations for the items. That is, ratings given by the users, textual representations of the items, sets of tags assigned to each item, and a comprehensive ontological organization of the tags. In this subsection, we describe a dataset satisfying these requirements that we used in our experiments.

#### 2.3.1. The Star Trek Television Series

Star Trek is a science fiction franchise that has influenced popular culture for more than 50 years [10], and remains a favorite among sci-fi enthusiasts the world over (STARFLEET, The International Star Trek Fan Association, Inc. (2019) [11]). The Star Trek media franchise canon boasts eight television series to date. Table 1 shows an overview. The episodes from the series TOS, TAS, TNG, and Voyager are used to test the various RSs proposed in this paper.

**Table 1.** The Star Trek television series overview.

| Series Title | Short Name | Original Release | No. of Seasons | No. of Episodes |
|---|---|---|---|---|
| Star Trek: The Original Series | TOS | 1966–1969 | 3 | 79 |
| Star Trek: The Animated Series | TAS | 1973–1974 | 2 | 22 |
| Star Trek: The Next, Generation | TNG | 1987–1994 | 7 | 178 |
| Star Trek: Deep Space Nine | DS9 | 1993–1999 | 7 | 177 |
| Star Trek: Voyager | Voyager | 1995–2001 | 7 | 172 |
| Star Trek: Enterprise | Enterprise | 2001–2005 | 4 | 99 |
| Star Trek: Discovery | Discovery | 2017–present | 2 | 29 |
| Star Trek: Shorts | Shorts | 2018–present | 1 | 4 |

#### 2.3.2. The Literary Theme Ontology

In this subsection, we describe version 0.1.3 of the Literary Theme Ontology (LTO) [13]. It is a controlled vocabulary of 2130 unique defined literary themes, hierarchically arranged into a tree structure according to the *is-a* relation. The maximum depth of the hierarchy is 7 and the maximum path length between any pair of entities is 13. LTO upper-level organization is inspired by a traditional classification system proposed by literary critic William Henry Hudson [13,52]. There are four upper-level theme domains:

| The Human Condition: | Themes pertaining to the inner and outer experiences of individuals be they about private life or pair and group interactions with others around them. |
| Society: | Themes pertaining to individuals involved in persistent social interaction, or a large social group sharing the same geographical or social territory, typically subject to the same political authority and dominant cultural expectations. These are themes about the interactions and patterns of relationships within or between different societies. |
| The Pursuit of Knowledge: | Themes pertaining to the expression of a view about how the world of nature operates, and how humans fit in relation to it. Put another way, these are themes about scientific, religious, philosophical, artistic, and humanist views on the nature of reality. |
| Alternate Reality: | Themes related to subject matter falling outside of reality as it is presently understood. This includes, but is not limited to, science fiction, fantasy, superhero fiction, science fantasy, horror, utopian and dystopian fiction, supernatural fiction as well as combinations thereof. |

Figure 2 depicts a bird's eye view of the ontology. The abstract theme "literary thematic entity" is taken as root class. Each domain is structured as a tree descended from the root with "the human condition", "society", "the pursuit of knowledge", and "alternate reality" serving as the top themes of their respective domains. Table 2 provides a summary of the number of classes (i.e., literary themes) in each domain and their heights from the root LTO class.

LTO is engineered to fit within the Basic Formal Ontology (BFO) top level ontology class hierarchy [53] in order to facilitate interoperability with other emerging fiction studies ontologies [54–58]. LTO is meant to cover important, operationally verifiable literary themes that can be expected reoccur in multiple works of fiction [13]. In designing LTO, we strove to make sibling classes mutually exclusive, but not necessarily jointly exhaustive. All literary themes are accompanied with definitions, and references when possible. We appealed to the principle of falsifiability in definition writing. That is to say, a well-defined literary theme will be such that it is possible to appeal to the definition to show it is not featured in a story. Take "the quest for immortality" as an example, which is defined as "A character is tempted by a perceived chance to live on well beyond what is considered to be a normal lifespan". The theme "the desire for vengeance" (Definition: A character seeks retribution over a perceived injury or wrong.) constitutes another example. By insisting on maximally unambiguous theme definitions, we aim to help bring the conversation of whether a theme is featured in a given work of fiction into the realm of rational argumentation. However, we fully acknowledge that the identification of literary themes in stories will always carry with it a certain element of subjectivity. It is the goal of LTO to minimize the subjective element in theme identification. The individual classes populating LTO at the early stage of development presented in this paper were mainly collected by watching Star Trek TOS, TAS, TNG, and Voyager episodes and recording the themes. We selected Star Trek for building up the ontology on account that the television series are culturally significant and explore a broad range of literary themes relating to the human condition, societal issues, as well as classic science fiction. That said, the ontology is admittedly science fiction oriented. In fact, an earlier version of LTO was used for the purpose of identifying over-represented themes in Star Trek series [59]. The later version of LTO (version 1.0.0) presented in Sheridan et al. [13] is populated with literary themes derived from a more robust collection of science fiction television series and films, including all the Star Trek television series shown in Table 1 save for *Discovery* and *Shorts*.

LTO was encoded using Web Ontology Language (OWL2) [60] and is made available for download at the project's GitHub repository (https://github.com/theme-ontology/theming) under a Creative Commons Attribution 4.0 International license (CC BY 4.0). It has also been made accessible in a structured manner through the R package **stoRy** [12]. This paper uses version 0.1.3 of the ontology, which can be accessed through the like versioned **stoRy** package 0.1.3. Functions for exploring the ontology are described in the package reference manual. For example, the command `theme$print()` prints summary information for the theme object `theme`, and the function `print_tree` takes a theme

object as input and prints the corresponding theme together with its descendants in tree format to the console. We encourage non-technical users to explore the current developmental version of the ontology on the Theme Ontology website (Theme Ontology (2019). URL: https://www.themeontology. org. (Online; accessed 30 June 2019)).

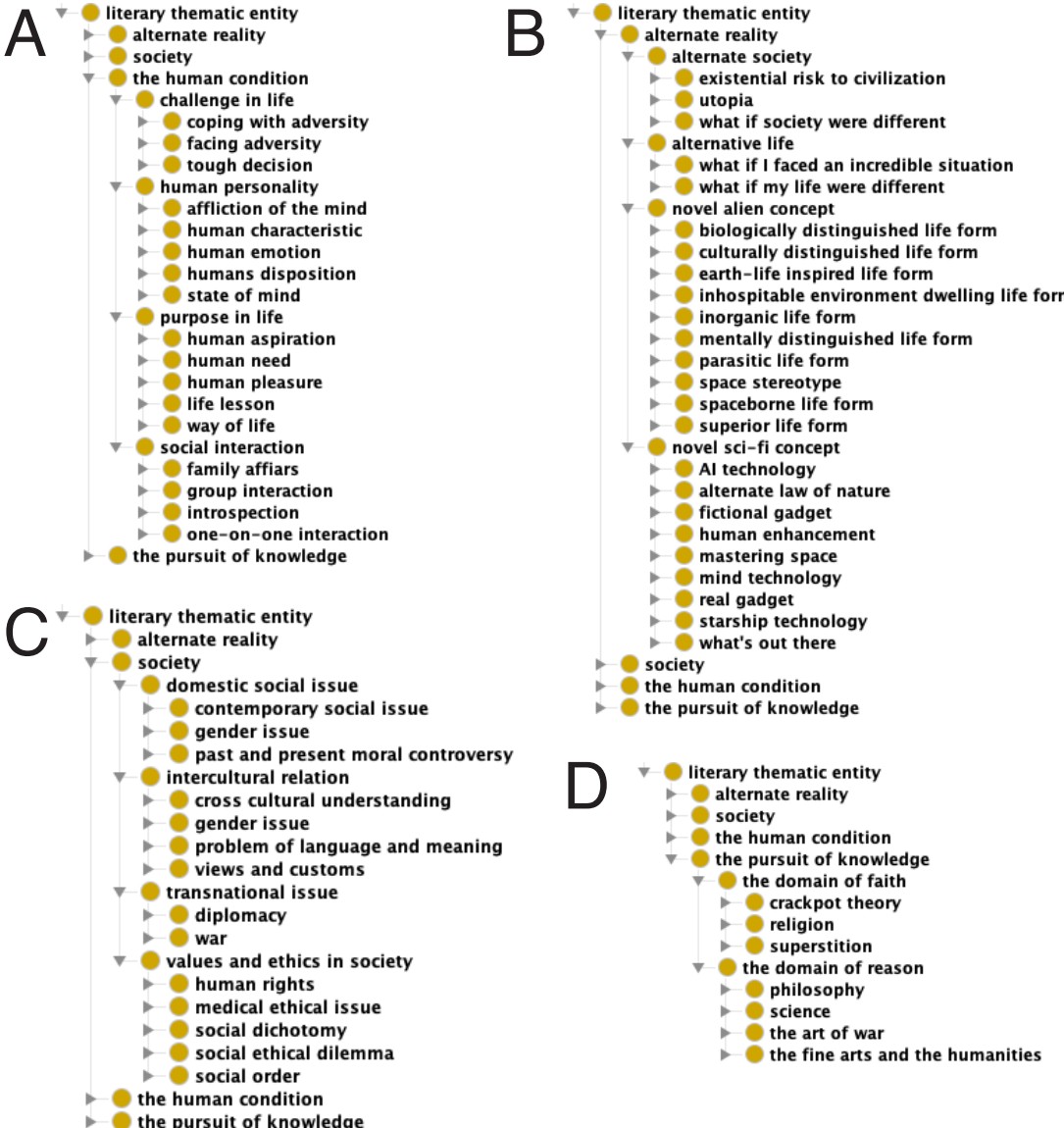

**Figure 2.** Literary Theme Ontology v0.1.3 class hierarchy overview. (**A–D**) show "the human condition", "alternate reality", "society", and "the pursuit of knowledge" domain themes to three levels of depth, respectively.

**Table 2.** Literary Theme Ontology v0.1.3 summary statistics.

| Domain Root Theme | Domain Color-Code | Theme Count | Leaf Theme Count | Tree Height |
|---|---|---|---|---|
| the human condition | 🔴 | 892 | 835 | 6 |
| society | 🟢 | 387 | 362 | 4 |
| the pursuit of knowledge | 🔵 | 329 | 308 | 4 |
| alternate reality | 🟡 | 521 | 484 | 4 |

2.3.3. A Thematically Annotated Star Trek Episode Dataset

We manually tagged a total of 452 Star Trek television series episodes with themes drawn from LTO v0.1.3. This covers all TOS, TAS, TNG, and Voyager television series episodes. Table 3 shows a basic statistical summary of the dataset. We distinguish between central themes (i.e., themes found to recur throughout a major part of a story or are otherwise important to its conclusion) and peripheral themes (i.e., briefly featured themes that are not necessarily part of the main story narrative).

**Table 3.** Thematically annotated Star Trek television episode summary statistics by series.

| Series Short Name | No. of Episodes | Mean Number of Central Themes per Episode $\pm$ S.D. | Mean Number of Peripheral Themes per Episode $\pm$ S.D. |
|---|---|---|---|
| TOS | 80 | 12.42 $\pm$ 4.31 | 20.05 $\pm$ 6.23 |
| TAS | 22 | 6.77 $\pm$ 2.58 | 3.41 $\pm$ 2.28 |
| TNG | 178 | 11.64 $\pm$ 4.38 | 14.88 $\pm$ 5.60 |
| Voyager | 172 | 9.20 $\pm$ 2.99 | 7.63 $\pm$ 3.38 |

In Appendix A, we provide a look at an example episode to illustrate the system of thematic annotation we employ. We recorded themes for each of the 452 episodes of TOS, TAS, TNG, and Voyager in a similar manner. The basic procedure we used in assigning themes is summed up as follows. We individually tagged episodes with themes before comparing notes with a view toward building a consensus set of themes for each episode. We aimed to abide in the principle of low-hanging fruit in the compilation of consensus themes. In the present context, this means we aimed to capture the more striking topics featured in each episode with appropriate themes. Another guiding principle is the minimization of false positives (i.e., the tagging of episodes with themes that are not featured) at the expense of tolerating false negatives (i.e., neglecting to tag episodes with themes that they feature). This "when in doubt, leave it out" strategy promotes erring on the side of caution.

*2.4. Episode Transcripts*

Episode transcripts for the series listed in Table 3 were extracted from a website maintained by fans of the Star Trek franchise (Transcripts available at Chrissie's Transcripts Site (2019) [61]). These transcripts contain the complete dialogues of the characters, brief descriptions of some actions, and captain and other officers' logs.

The texts were preprocessed by removing stop-words, punctuation marks, and words occurring only in one episode transcript. The practical motivation for this is that stop-words tend to occur statistically similar in any piece of text in English making their analysis non-informative for text discrimination. Similarly, unique words do not contribute to the analysis of commonalities between texts. In addition, we reduced the words to their stems by using the Porter's stemmer [62]. Given the relatively small number of episode transcripts, the process of stemming contributes to reducing the size of the vocabulary used in the collection, and therefore it contributes to reducing the sparsity of the representation. The size of the resulting vocabulary is 14,262, the average number of words (stems) per episode is 2916 with a standard deviation of 801; in addition, 6179 is the maximum number of words in an episode, and 1301 is the minimum. All preprocessing tasks were performed using the Natural Language Toolkit (NLTK) [63].

*2.5. User Preferences*

User and ratings data for the Star Trek television series episodes were obtained from the user reviews on the Internet Movie Database (IMDb) (For an example review page for the episode annotated in Appendix A see [64]). Each rating given by a user to an episode is on a preference scale from 1 and 10 stars. The extraction produced a dataset of 3975 ratings given by 842 users to 396 episodes. Figure 3

shows the distribution of the ratings. This dataset was collected in December 2018 and contains all ratings available for the Star Trek episodes in IMDb up to that time.

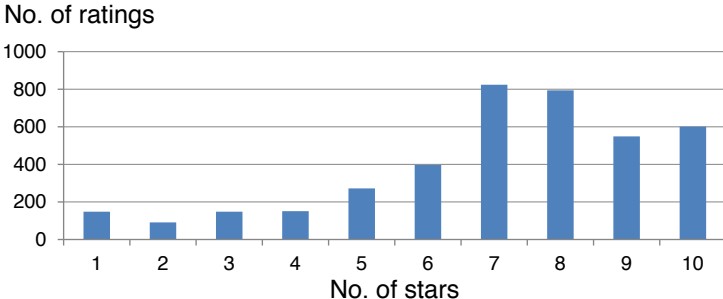

**Figure 3.** Star Trek television series episode IMDb user ratings distribution as of December 2018.

## 3. Experimental Validation

The goal of this experimental validation is to compare the performance of RSs built with different types of resources using as a testbed a single set of items to be recommended (i.e., Star Trek episodes). In short, we used four information sources to leverage the RS engines: (1) CBF using item content (i.e., transcripts), (2) KBF using items tagged with a set of labels from a controlled vocabulary (i.e., LTO themes), (3) OBF using knowledge in the form of an ontology (i.e., LTO themes with ontology structure), and (4) CF using user preferences (i.e., ratings). Aside from the classical rating prediction task (Warm System), we evaluated the methods in the item cold-start scenario (Cold Start) and analyzed their parameters.

### 3.1. Experimental Setup

We carried out the experiments using the MyMediaLite RS library [65], which includes an implementation of the IKNN model [14] along with a host of other classical RS algorithms [15,66–70]. To test the item-similarity functions presented in Section 2.2, we predicted user ratings for the Star Trek television series episode data presented in Section 2.5. For evaluation, the ratings in the data were randomly divided into two parts: 70% for training the model and 30% for testing it. For the item cold-start scenario, the training-test partitions were drawn by splitting the items (not the ratings) and selecting those partitions covering roughly 30% of the total ratings in the test partition. The selected performance measure is the root-mean-square error (RMSE) as measured between the actual ratings and the predictions produced by the RS method in the test partition. Next, we produced 30 shuffles of such partitions and evaluated the RMSE of each method for each partition. The final reported result for each model is the average RMSE across the 30 training-test partitions. We used the Wilcoxon rank-sum test statistic to assess the statistical significance of the observed differences in performance between methods.

The RS methods to be tested are classified into four groups according to the resources they use. The first three categories are variations of the IKNN method presented in Section 2.1, in which we substitute the $s_{i,j}$ of Equation (1) by the different item similarity functions presented in Section 2.2. In practice, each variation produces a text file containing on each line the identifiers of two items and their corresponding similarity score. Then, the MyMediaLite application is instructed to use this file as source for computing the similarities between items. For the sake of a fair comparison, the system ($\mu$), item ($b_i$), and user ($b_i$) biases (see Equation (2)) are used in all the IKNN algorithm variations tested. The fourth category comprises an assortment of CF methods and baselines that are used to ensure for a comprehensive comparison.

Below, we present a summary of the methods used in the experiments:

CBF recommenders using transcripts: These methods use the data and preprocessing procedure described in Section 2.4. TFIDF is implemented using Equations (3) and (4), and LSI is implemented by

performing SVD, as described in Section 2.2.1, on the document-term matrix obtained from the data. The number of latent factors was varied from 10 to 100 in increments of 10. Both approaches are implemented using the Gensim (Gensim: Topic modelling for humans (2019). URL: https://radimrehurek.com/gensim/. (Online; accessed 30 June 2019)) text processing library [71].

**KBF recommenders using themes:** The three methods described in Section 2.2.2 applied to the thematically annotated representation of the episodes described in Section 2.3.3. JACCARD and DICE were implemented using the Equation (5) formulae as item similarity functions, while COSINE_IDF was implemented using Equation (6).

**OBF recommenders using themes and the ontology:** This category comprises the methods introduced in this work, which are described in detail in Section 2.2.3. These methods make use of the thematically annotated Star Trek episodes described in Section 2.3.3, and of the LTO themes as presented in Section 2.3.2. Each of the six variants is named after the abbreviation for their assocaited item similarity function: PATH, WUP, RES, LCH, LIN, and JCN.

**CF recommenders and baselines:** In this group, we tested a set of classical RSs based purely on user ratings. These methods can be grouped into KNN [14] and matrix factorization approaches [15,66–70]. In addition, we included five popular baseline methods: (1) User Item Baseline, which produces rating predictions using the baselines described in Section 2.1; (2) Item Average Baseline, which uses as predictions the mean rating of each item; (3) User Average Baseline same as before, but averaging by user; (4) Global Average Baseline which always predicts the average rating of the dataset; and (5) Random Baseline, which produces random ratings distributed uniformly.

### 3.2. Results

Figure 4 shows the results obtained by the methods presented in this work based on IKNN. For each method, the parameter *k* (the number of nearest neighbors) was ranged from 10 to 100 in increments of 10. As the performance measure RMSE is an error measure, the lower the score, the better the performance of the tested method. In addition, as usual with star-scaled rating prediction experiments, the differences between methods are only noticeable in the second and third decimal places.

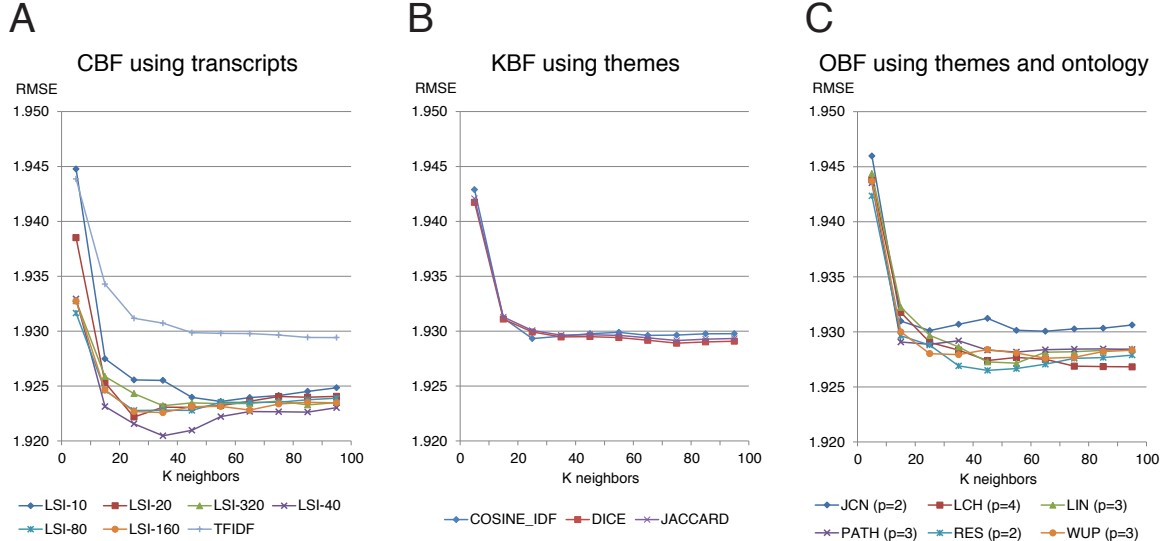

**Figure 4.** Results for the RSs (recommender systems) built using transcripts, themes, and the ontology of themes. (**A**) CBF using transcripts; (**B**) KBF using themes; (**C**) OBF using themes and ontology.

Figure 4A shows the results for the CBF recommenders, which obtained the best results. In particular, the best configuration is LSI-40 (40 latent factors) with *k* = 40. The other LSI methods

performed similarly. TFIDF produced the worst result out of the CBF recommenders. Figure 4B shows the KBF recommender results. The three methods performed practically identically with DICE showing a slight advantage over the others. Finally, Figure 4C shows the results obtained using the novel OBF recommenders, which performed in between the first two groups. Since these methods depend on the softness control parameter $p$, the figure shows the results using the best value of $p$ for each method. Figure 5 shows how this parameter behaves for each one of the methods. The LIN, WUP, and LCH methods clearly exhibit robust behavior regarding that parameter.

In Table 4, we compare the best performing CBF, KBF, and OBF configurations (the first four rows) shown in Figure 4 with a variety of CF recommenders and baselines. We report the resources used for building each model and their average RMSE values with standard deviation. In the first column, we labeled the top six methods numerically from 1 to 6 in order to report pairwise hypothesis test results in the last six columns. If the null hypothesis of equal performance is rejected for two methods being compared, then we record an "*" mark (for clarity we record an "=" when the table entry corresponds to the same method). Since the numerical differences in the observed RMSE values are narrow, we increased the typical statistical significance level from $p < 0.05$ to $p < 0.01$. Note that the number of paired samples for each test corresponds to the 30 random partitions of the data. The first set of columns corresponds to the results and hypothesis testing for the *Warm System* scenario, while the second set of columns corresponds to those for the *Cold Start* scenario.

**Table 4.** Performance comparison between the best IKNN (item K-nearest neighbors) models and other CF (collaborative filtering) approaches in the Warm System and Cold Start evaluation settings. Method #1 uses transcripts and ratings. Methods #2 and #3 use themes, ontology, and ratings. Method #4 uses themes and ratings. The remaining methods use only ratings with the exception of the random baseline method. Note that the "*" mark is used to indicate that method performance is significantly different at significance level 0.01.

| # | Type | Method Description | Warm System Scenario | | | | | | | Cold Start Scenario | | | | | | |
|---|---|---|---|---|---|---|---|---|---|---|---|---|---|---|---|---|
| | | | RMSE ± s.d. | 1 | 2 | 3 | 4 | 5 | 6 | RMSE ± s.d. | 1 | 2 | 3 | 4 | 5 | 6 |
| 1 | CBF | IKNN-LSI-40, k = 40 [this paper] | 1.920 ± 0.037 | = | * | * | | * | * | 1.466 ± 0.685 | = | * | * | | | * |
| 2 | OBF | IKNN-RES, p = 2, k = 50 [this paper] | 1.927 ± 0.040 | * | = | | | * | * | 1.583 ± 0.733 | * | = | | | | * |
| 3 | OBF | IKNN-LCH, p = 4, k = 80 [this paper] | 1.927 ± 0.040 | * | | = | | * | * | 1.596 ± 0.747 | * | | = | | | * |
| 4 | KBF | IKNN-DICE, k = 70 [this paper] | 1.929 ± 0.046 | | | | = | | * | 1.601 ± 0.786 | | | | = | | * |
| 5 | CF | IKNN, k = 40 [14] | 1.940 ± 0.039 | * | * | * | | = | * | 1.597 ± 0.746 | | | | | = | * |
| 6 | CF | Sig. Item Asymm. FM, f = 10 [66] | 1.977 ± 0.038 | * | * | * | * | * | = | 1.763 ± 0.848 | * | * | * | * | * | = |
| | CF | Sig. Comb. Asymm. FM, f = 10 [66] | 1.978 ± 0.040 | * | * | * | * | * | | 1.783 ± 0.842 | * | * | * | * | * | |
| | CF | Sig. User Asymm. FM, f = 10 [66] | 1.982 ± 0.039 | * | * | * | * | * | | 1.809 ± 0.788 | * | * | * | * | * | |
| | CF | User Item Baseline [14] | 2.007 ± 0.040 | * | * | * | * | * | * | 1.753 ± 0.851 | * | * | * | * | * | |
| | CF | User KNN, k = 80 [14] | 2.021 ± 0.041 | * | * | * | * | * | * | 1.753 ± 0.851 | * | * | * | * | * | |
| | CF | Biased MF, f = 10 [68] | 2.084 ± 0.033 | * | * | * | * | * | * | 1.767 ± 0.905 | * | * | * | * | * | |
| | CF | SVD Plus Plus, f = 10 [69] | 2.120 ± 0.043 | * | * | * | * | * | * | 1.755 ± 0.842 | * | * | * | * | * | |
| | CF | Slope One [70] | 2.121 ± 0.039 | * | * | * | * | * | * | 1.835 ± 0.817 | * | * | * | * | * | |
| | CF | Item Average Baseline | 2.164 ± 0.045 | * | * | * | * | * | * | 1.835 ± 0.817 | * | * | * | * | * | |
| | CF | User Average Baseline | 2.224 ± 0.043 | * | * | * | * | * | * | 1.774 ± 0.870 | * | * | * | * | * | |
| | CF | MF, f = 10 [15] | 2.254 ± 0.044 | * | * | * | * | * | * | 1.835 ± 0.817 | * | * | * | * | * | |
| | CF | Global Average Baseline | 2.320 ± 0.040 | * | * | * | * | * | * | 1.835 ± 0.817 | * | * | * | * | * | |
| | CF | Factor Wise MF, f = 10 [67] | 2.787 ± 0.148 | * | * | * | * | * | * | 1.822 ± 0.881 | * | * | * | * | * | |
| | | Random Baseline | 3.857 ± 0.057 | * | * | * | * | * | * | 3.626 ± 1.276 | * | * | * | * | * | * |

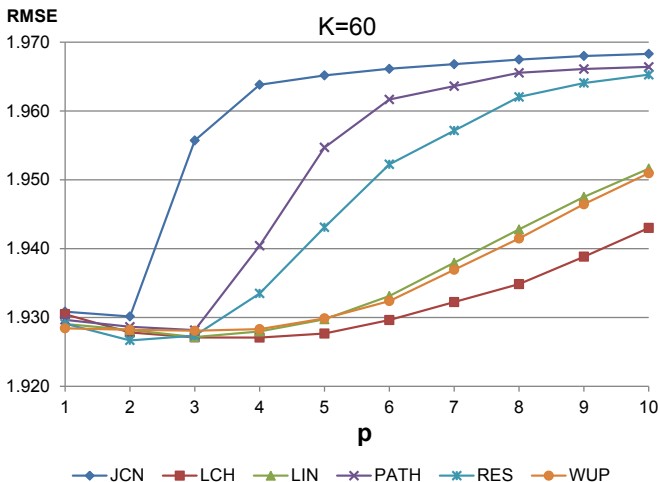

**Figure 5.** Behavior of softness control parameter $p$ from Equation (8) in the OBF recommenders when the number of nearest neighbors $k$ is fixed at 60.

### 3.3. Results Discussion

Let us first discuss the results for the Warm System scenario. The best results were achieved by the LSI method. In contrast, among CBF recommenders, TFIDF performed considerably poorer, meaning that the sparsity of this representation hinders its ability to model the items (see Figure 4A). The optimal number of latent factors is 40, which must be assessed against the original vocabulary size (~14 K words) and the average transcript length (~3 K words). Other domains having larger vocabularies and textual representations can be expected to require more latent factors and vice versa.

The KBF recommenders based on themes (COSIDE_IDF, DICE, and JACCARD) performed practically identically to TFIDF, meaning that the effort involved in representing the items using a controlled vocabulary of size ~2 K did not provide much benefit. In spite of this finding, all the methods that used thematic representation in combination with the ontology managed to improve the results (with the exception of the JCN method). Therefore, the knowledge encoded in the ontology and the novel method for exploiting the ontology are useful in spite of the sparsity of the representation that was used. In addition, this result suggests that even a representation of relatively moderate dimensionality (~2 K entities) could benefit from a combination with a method such as LSI.

Among the OBF recommenders, RES and LCH tied for the best performance. However, Figure 5 shows that RES is rather sensitive to the parameter $p$, while LCH exhibited the best robustness on that matter. In addition, RES requires IC, which also requires the existence of a relatively large corpus to compute those statistics.

The graphs in Figure 4 also show performance variation due to parameter $k$, which is the number of nearest neighbors used in IKNN. The clear tendency is to get lower error rates as $k$ increases. There is a general inflection point around $k = 40$ where more neighbors provide only a small benefit, and even an increased RMSE beyond this point for two of the best performing methods: LSI-40 and RES.

Regarding the comparison of CBF/KBF/OBF recommenders against their CF counterparts, we observed that only the KBF recommenders fail to significantly outperform the CF ones in the Warm System scenario. It is important to note that, in that scenario, our modifications of the IKNN algorithm using LSI-40 and LCH do outperform the classical purely collaborative IKNN and remaining CF tested approaches. The most notable difference in the results of the Cold Start scenario is that all methods exhibit comparatively large variances across the 30 random partitions. Although all the IKNN variants manage to surpass all other CF recommenders, the latter do not produce significant statistical differences among them. Out of the IKNN approaches, the only statistically significant difference was observed between the CBF recommender (LSI-40) and the OBF recommenders (RES and LCH).

The results also show that the CBF and OBF recommenders behave similarly in comparison to the other approaches, but the CBF recommender consistently outperforms the OBF one by a small margin. This difference is attributable to the particularly good quality of the content-based representation in the cause of our dataset. The fan curated Star Trek episode transcripts provide high-quality and detailed descriptions that even include fragments without dialog. This particular situation leads us to conclude that the effort invested in the construction of a detailed ontology is not worth it for recommendation purposes. However, that kind of representation is only possible for certain types of items (e.g., books, movies, TV shows) and, even if that is possible, its availability is not guaranteed. In our opinion, it is possible that the cost and effort involved in providing high-quality textual descriptions of the items surpass those of building a detailed domain ontology.

In conclusion, LCH using $p = 4$ and $k = 80$ is a reasonable choice when a relatively large ontology is available, and LSI-40 is a good choice when an appropriate textual representation of the items is available. In the event that none of these resources are available, then DICE can be expected to have a performance equivalent to the classic IKKN algorithm.

## 4. Discussion

In this paper, we presented a novel set of OBF recommenders aimed at mitigating the item cold-start problem in the domain of fiction content recommendation. Unlike most conventional approaches [3,22,24,25], which exploit lightweight ontological representations of users and items, we propose a scheme for factoring large taxonomic hierarchies of item features directly into the recommendation process. In a case study, we compared the performance of our proposed OBF recommenders against a variety of alternatives in a Star Trek television series episode user rating prediction exercise. For comparison's sake, we implemented: (1) conventional CF recommenders based solely on user ratings, (2) CBF recommenders based on episode transcripts using traditional text mining practices, and (3) novel KBF recommenders based on LTO thematically annotated Star Trek episodes without the ontology hierarchy. Meanwhile, the OBF recommenders exploited LTO thematically annotated Star Trek episodes together with LTO ontology hierarchy. We found the CBF, KBF and OBF approaches to be suitable alternatives to CF for the Cold Start stage of an RS's lifetime. The CBF approach obtained the best results, suggesting that it is to be preferred over the KBF/OBF alternatives when an informative textual representation of the items under consideration is available. However, interestingly, we found that the OBF approach outperformed the KBF one, indicating that the ontology hierarchy is informative above and beyond the terms that populate it. This result provides definitive evidence in support for our original research hypothesis. However, the CF approach is the clear choice in the case of the Warm System stage of an RS when there is already a continuous supply of explicit or implicit user preferences.

We conclude based on our experimental evaluation that the best approaches tested in each of the groups of algorithms (CBF, KBF, OBF, and CF) performed similarly with some statistically significant differences but with overall small effect sizes. This result should be evaluated taking into consideration that the dataset used was exceptionally balanced in the quality of the information resources that each approach exploited. For instance, the used content-based representation in the form of episode transcripts contained particularly compact and informative semantic descriptions of the items, which is not commonly available for other domains. Since the CBF recommenders tested obtained the best results, practitioners should transfer this finding to other domains with caution, if the content representations are not as informative or detailed as the ones used in this study. Similarly, regarding the thematic annotation of the episodes (KBF approach), our methodology was rather thorough involving a significant manual effort. Again, we suggest that practitioners should compare their methodologies of item annotation with the example provided in Appendix A to put our results in context. Finally, the same remark applies to the results of the OBF recommenders, since the ontology used was of considerable size and depth. Therefore, we recommend that, in order to transfer the

results to other domains, the characteristics of the ontologies used ought to be compared with the description of Section 2.3.2 (also see [12,13]).

The perspectives of future work that are opened from this study are diverse. In our opinion, the main path is to take advantage of the proposed Star Trek television series testbed for the development of multiple-modality hybrid recommendation algorithms. The inclusion of users reviews constitutes an interesting extension of our testbed. Doing so would effectively extend the analysis to other affective dimensions that can be extracted from the reviews, given recent developments in natural language processing using deep learning applied to sentiment analysis [72]. Additionally, we consider that the set of aligned data resources offers the opportunity to explore new tasks in the area of artificial intelligence. For example, the alignment between transcripts and thematic annotations is an interesting input for an automatic annotation approach based on machine learning. Likewise, the modality of user preferences introduces an interesting factor that opens up the possibility of developing algorithms for the automatic creation of personalized literary content.

**Supplementary Materials:** The Literary Theme Ontology can be found at https://github.com/theme-ontology/theming. The code and other resources to reproduce the experiments can be found at https://github.com/sgjimenezv/star_trek_recsys. The R Shiny web application code and related files can be found at https://github.com/theme-ontology/shiny-apps.

**Author Contributions:** The authors contributed equally to this work.

**Funding:** This research received no external funding.

**Acknowledgments:** We kindly thank Jose A. Dianes and Oshan Modi for coding significant portions of the R **Shiny** web application, and Takuro Iwane for preparing various R scripts.

**Conflicts of Interest:** The authors declare no conflict of interest.

## Abbreviations

The following abbreviations are used in this manuscript:

| | |
|---|---|
| BFO | Basic Formal Ontology |
| CF | collaborative filtering (general approach for recommender systems) |
| CBF | content-based filtering (general approach for recommender systems) |
| DF | document frequency |
| FM | factor model (general approach for CF recommender systems) |
| IC | information content |
| IDF | inverse document frequency |
| IKNN | item K-nearest neighbors (method for recommender systems) |
| IMDb | The Internet Movie Database |
| JCH | Jiang and Conrath measure [73] (entity similarity function in an ontology hierarchy) |
| KBF | knowledge-based filtering (general approach for recommender systems) |
| LCH | Leacock and Chodorow measure [47] (entity similarity function in an ontology hierarchy) |
| LCS | least common subsumer between two entities in an ontology |
| LIN | Lin's measure [49] |
| LSI | latent semantic indexing (method for text representation) |
| LTO | Literary Theme Ontology |
| MF | matrix factorization |
| NLTK | Natural Language Toolkit |
| OBF | ontology-based filtering (general approach for recommender systems) |
| RES | Resnik's measure [46] (entity similarity function in an ontology hierarchy) |
| RMSE | root-mean square error |
| RS | recommender system |
| SVD | singular value decomposition |
| s.d. | standard deviation |
| TAS | Star Trek: The Animated Series (series of Star Trek TV episodes) |
| TF | term frequency |
| TFIDF | term frequency–inverse document frequency (method for text representation) |

TNG　　Star Trek: The Next, Generation (series of Star Trek TV episodes)
TOS　　Star Trek: The Original Series (series of Star Trek TV episodes)
WUP　　Wu and Palmer's measure [48] (entity similarity function in an ontology hierarchy)

## Appendix A

Table A1 catalogs the literary themes for the Voyager episode False Profits (1996). In this episode, the USS Voyager starship crew discover a planet on which two Ferengi, named Arridor and Kol, have duped the comparatively primitive inhabitants thereof into thinking them holy prophets. The story begins with Commander Chakotay and Lieutenant Tom Paris beaming down to the Takarian homeworld to investigate signs of "matter replicator" usage among the local inhabitants. This is considered odd because the Takarians otherwise manifest only a Bronze Age level of technology. Chakotay and Paris soon uncover how the Ferengi had traveled through a "wormhole", crash-landed on the planet, and, in a naked display of "science as magic to the primitive", convinced the Takarians that they had come in "fulfillment of prophesy" through the performance of "matter replicator" powered conjuring tricks. The intervening years saw the Ferengi use "religion as a control mechanism" to shape the Takarian economy to suit their own self-interest. Arridor and Kol now wallow in the muck of "avarice" as a result of their "fraud". Back aboard the ship, Captain Kathryn Janeway confers with her senior staff about "the ethics of interfering in less advanced societies", before venturing to determine a proper course of action. Janeway decides that this appalling "exploitation of sentient beings" must be brought to an end with minimal interruption to the internal development of Takarian civilization. Because forcibly removing Arridor and Kol could undermine Takarian religion, she reasons that the pair must be made to leave the planet of their own accord. Morale Officer Neelix, disguised as a representative of the Ferengi head of state, beams down to the planet in an effort to dupe Arridor and Kol into returning to their homeworld. However, the Ferengi, driven by an insatiable "lust for gold", refuse to leave without putting up a fight. The situation quickly spirals out of control when the Takarians opt to burn Arridor, Kol, and Neelix at the stake. This, according to their "primitive point of view", would deliver the holy prophets back to the heavens from whence they came. Arridor and Kol resort to blatant "casuistry in interpretation of scripture" in a last ditched attempt to save their skins, but to no avail. Then, at the very moment when all hope seems lost, the condemned men are beamed up to the USS Voyager just as the smoke begins to overwhelm, as the Takarian onlookers watch in amazement at the return of their holy prophets to the stars.

**Table A1.** Inventory of the Star Trek: Voyager episode False Profits (1996) themes. The domain color-codes are red for "the human condition", green for "society", yellow for "alternate reality", and blue for "the pursuit of knowledge".

| Literary Theme | Domain | Level | Comment |
|---|---|---|---|
| avarice | 🔴 | central | Arridor and Kol exploit a Bronze Age people for economic gain. |
| exploitation of sentient beings | 🟢 | central | Arridor and Kol exploit a Bronze Age people for economic gain. |
| fraud | 🟢 | central | Arridor and Kol fraudulently claim to be the Holy Sages prophesied in Takarian sacred scripture. |
| primitive point of view | 🟢 | central | The viewer is made to see the world through the eyes of a Bronze Age people. |
| religion as a control mechanism | 🟢 | central | Arridor and Kol use religion as a means of exploiting a technologically lesser advanced people. |
| science as magic to the primitive | 🟢 | central | Arridor and Kol use advanced technology to trick a Bronze Age people into thinking them gods. |

**Table A1.** *Cont.*

| Literary Theme | Domain | Level | Comment |
|---|---|---|---|
| the ethics of interfering in less advanced societies | 🟢 | central | Captain Janeway argued she had the authority to depose Arridor and Kol from their seat of power on the Takarian homeworld because the Federation was responsible for the cultural contamination caused by their arrival. |
| the fulfillment of prophesy | 🔵 | central | Arridor and Kol fraudulent claim to be the Holy Sages prophesied in Takarian sacred scripture. |
| the lust for gold | 🔴 | central | Arridor and Kol exploit a Bronze Age people for economic gain. |
| casuistry in interpretation of scripture | 🔵 | peripheral | Arridor and Kol advocated a nonliteral interpretation of the passage in Takarian sacred scripture condemning them to being burned at the stake. |
| wormhole | 🟡 | peripheral | Arridor and Kol travel through a wormhole to reach the Takarian homeworld. |
| matter replicator | 🟡 | peripheral | Arridor and Kol proliferated matter replicator technology on the Takarian homeworld. |

**Appendix B**

Here, we show how our R Shiny web application can be employed to recommend Star Trek television series episodes that are similar to the Voyager episode False Profits (1996; story ID = *voy3x05*). A synopsis of the False Profits episode is provided in Appendix A, and an inventory of the literary themes featured therein is found in Table A1.

Figure A1 shows a screenshot of the R Shiny web application in action. Viewing the said screenshot, it is easy to imagine a hypothetical user, who, having selected False Profits from a dropdown menu of episodes, peruses the returned table of recommended Star Trek episodes with marked delight. Our hypothetical user has elected to use the cosine index in an effort to find similar stories on the basis of shared central themes. The file StarTrek.smt, which contains Star Trek episode story IDs, they have uploaded as a background storyset so as to restrict the recommendations to the 452 Star Trek episodes considered in this work. Note that the R Shiny web application code and StarTrek.smt file are available for download at https://github.com/theme-ontology/shiny-apps.

Now on to the recommendations. The most similar episode to False Profits is revealed to be Devil's Due (1991, story ID = *tng4x13*). The TNG classic shares six central themes with its Voyager counterpart in all. In the story, a woman claiming to be the devil of Ventaxian mythology returns to enslave the people of Ventax II in accordance with an ancient contract. However, Captain Picard is convinced she is nothing more than an opportunistic charlatan. If our user can somehow restrain themself from watching Devil's Due straightaway, they will no doubt be pleased to notice a combination of "avarice/the lust for gold" and "the ethics of interfering in less advanced societies" featured in the three subsequent recommendations. It is interesting to note that, unlike with Devil's Due, these three episodes do not touch on religion. Each of the remaining top ten recommended episodes is related to False Profits by themes from exactly one of the domains of the human condition (e.g., "avarice" and "the lust for gold"), society (e.g., "the ethics of interfering in less advanced societies"), and the pursuit of knowledge (e.g., "religion as a control mechanism" and "the fulfillment of prophesy"). Thus, we see how our user, furnished with these recommendations, is well launched on the selection of their next episodes to watch.

## Story recommender

**Figure A1.** Story recommender R Shiny web application screenshot. The table lists the top ten most similar episodes to the Voyager episode False Profits. Pairwise episode similarity is determined by applying the cosine index to central themes. The file StarTrek.smt is a background storyset file containing story IDs for all 452 thematically annotated Star Trek television series episodes considered in this work.

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
