# Peer review of "An Ontology-Based Recommender System with an Application to the Star Trek Television Franchise"

_futureinternet, doi:10.3390/fi11090182_

Round 1

Reviewer 1 Report

The manuscript is well-written centered on an interesting topic. Organization of the paper is good and the proposed method is quite novel. The manuscript, however, does not link well with recent literature on sentiment analysis appeared in relevant top-tier journals, e.g., the IEEE Intelligent Systems department on "Affective Computing and Sentiment Analysis". Also, latest trends in deep learning based aspect extraction, e.g., attentive LSTM, are missing. Additionally, check recent relevant literature on word representations for sentiment analysis and on learning binary codes for efficient recommendation systems. Finally, some small issues with presentation (e.g., RSs produces) need to be fixed.

Author Response

Response to Reviewer 1 Comments

Point 1: Organization of the paper is good and the proposed method is quite novel. The manuscript, however, does not link well with recent literature on sentiment analysis appeared in relevant top-tier journals, e.g., the IEEE Intelligent Systems department on "Affective Computing and Sentiment Analysis".

Response 1: We added a sentence to the end of the first paragraph of the Introduction section, where we made the connection with the literature suggested by the reviewer. See lines 26-29.

Point 2: Also, latest trends in deep learning based aspect extraction, e.g., attentive LSTM, are missing.

Response 2: We linked a possible future work to the suggested literature in the last paragraph of the Discussion and Conclusion section. See lines 502-505.

Point 3: Additionally, check recent relevant literature on word representations for sentiment analysis and on learning binary codes for efficient recommendation systems.

Response 3: We cited the suggested literature in a new sentence in the Introduction section in the second sentence at the fifth paragraph. See lines 59-61.

Point 4: Finally, some small issues with presentation (e.g., RSs produces) need to be fixed.

Response 4: We have rewritten the first paragraph of the Introduction section. The "RSs produces" part of the text has been completely removed. See lines 23-29. In addition, we corrected a number of minor grammatical issues throughout the paper. Red text is used to show where the manuscript has been modified.

Reviewer 2 Report

* summary:
In this paper, the authors proposed a recommendation system for similar items. In the proposed method, the ontology-based similarity measure was employed in order to mitigate the item cold-start problem. As a testbed, Star Trek Television series was used to compare the proposed method with existing recommendation systems.

* review:
Ontology-based method is an interesting approach to improve the prediction accuracy of recommendation systems. Also, Star Trek data as a benchmark dataset is remarkable contribution of this research area. After minor revision, I'm willing to recommend the publication from the Future Internet journal.

minor comments:
Adding a clear exposition of "cold start" and "warm start" would be helpful to the readers.
In eq (1): in the right-hand side, r_{u,i} and b_{u,i} should be r_{u,j} and b_{u,j}, respectively.
Line 206: What is ORS?
In Table 4, CBF outperformed the other methods including OBF. A clear explanation of the numerical results would be great.

Author Response

Response to Reviewer 2 Comments

Point 1: Adding a clear exposition of "cold start" and "warm start" would be helpful to the readers. 

Response 1: We added a paragraph to the introduction addressing the reviewer suggestion. See lines 40-48.

Point 2: In eq (1): in the right-hand side, r_{u,i} and b_{u,i} should be r_{u,j} and b_{u,j}, respectively. 

Response 2: We corrected the equation as suggested by the reviewer.

Point 3: Line 206: What is ORS?

Response 3: The acronym ORS was mistakenly used to refer OBF, therefore it has been corrected in the text.

Point 4: In Table 4, CBF outperformed the other methods including OBF. A clear explanation of the numerical results would be great.

Response 4: We added a paragraph at the end of the Results Discussion subsection discussing that remark. See lines 450-459.

In addition to improving the paper by addressing the valuable comments of the reviewers, we proof-read the manuscript. Red text is used to show where the manuscript has been modified.